# Human Adipose-Derived Stem Cells Reduce Cellular Damage after Experimental Spinal Cord Injury in Rats

**DOI:** 10.3390/biomedicines11051394

**Published:** 2023-05-08

**Authors:** Emiliano Neves Vialle, Letícia Fracaro, Fabiane Barchiki, Alejandro Correa Dominguez, André de Oliveira Arruda, Marcia Olandoski, Paulo Roberto Slud Brofman, Carmen Lúcia Kuniyoshi Rebelatto

**Affiliations:** 1Spine Surgery Group, Cajuru University Hospital, Pontifícia Universidade Católica do Paraná, Curitiba 80215-030, Brazil; 2Core for Cell Technology, School of Medicine and Life Sciences, Pontifícia Universidade Católica do Paraná, Curitiba 80215-030, Brazil; 3National Institute of Science and Technology for Regenerative Medicine, INCT-REGENERA, Rio de Janeiro 21941-599, Brazil; 4Laboratory of Basic Biology of Stem Cells, Carlos Chagas Institute—Fiocruz, Rio de Janeiro 21941-599, Brazil; 5Department of Biostatistics, School of Medicine, Catholic University of Paraná, Curitiba 80215-030, Brazil

**Keywords:** spinal cord injury, cell therapy, adipose tissue, neurological rehabilitation, experimental surgery

## Abstract

Traumatic spinal cord injury (SCI) is a devastating condition without an effective therapy. Cellular therapies are among the promising treatment strategies. Adult stem cells, such as mesenchymal stem cells, are often used clinical research for their immunomodulatory and regenerative potential. This study aimed to evaluate the effect of human adipose tissue-derived stem cells (ADSC) infusion through the cauda equina in rats with SCI. The human ADSC from bariatric surgery was isolated, expanded, and characterized. Wistar rats were subjected to blunt SCI and were divided into four groups. Two experimental groups (EG): EG1 received one ADSC infusion after SCI, and EG2 received two infusions, the first one after SCI and the second infusion seven days after the injury. Control groups (CG1 and CG2) received infusion with a culture medium. In vivo, cell tracking was performed 48 h and seven days after ADSC infusion. The animals were followed up for 40 days after SCI, and immunohistochemical quantification of myelin, neurons, and astrocytes was performed. Cellular tracking showed cell migration towards the injury site. ADSC infusion significantly reduced neuronal loss, although it did not prevent the myelin loss or enhance the area occupied by astrocytes compared to the control group. The results were similar when comparing one or two cell infusions. The injection of ADSC distal to the injured area was shown to be a safe and effective method for cellular administration in spinal cord injury.

## 1. Introduction

Traumatic spinal cord injury (SCI) is responsible for high morbidity and mortality rates in our society, affecting a significant portion of the economically active population and causing significant socioeconomic impacts [1,2]. Regarding the pathophysiology of SCI, after the initial triggering factor of the injury, there is a complex and intricate sequence of events that results in neuronal death and functional loss below the affected level, with interruption of the circuits formed by the afferent sensory fibers and efferent motors that communicate between the cerebral cortex and the rest of the body [3,4]. Spinal cord injury is divided into two phases: primary injury—mechanical, acute, with direct injury and compression of neural elements; and secondary—ischemic, related to a reactive biochemical cascade with changes in perfusion and local blood flow [5,6,7]. Several therapeutic strategies have been proposed to treat or minimize the damage caused by SCI; surgical procedures are not able to act on the biochemical processes of secondary injury or provide stimulus to regeneration [5]. Other options of treatment are corticoids, riluzole, hormones, calcium channel blockers, and specific proteins, which are ineffective in the treatment of SCI [8,9]. The administration of stem cells (SC) presents as a promising strategy since it can potentially interrupt or minimize the cascade of pathophysiological events and provide a favorable environment for specialized tissue regeneration, with possible positive clinical correlation [10,11,12]. Adult stem cells, such as mesenchymal stem cells (MSC), are often used in clinical research for their immunomodulatory and regenerative potential [9,13]. The adipose tissue is an attractive source of MSC because of the high frequency of adipose-derived stem cells (ADSC); they present few ethical concerns and can be obtained from donors undergoing plastic surgical procedures [14,15]. Considering the complexity of SCI and the different mechanisms of action of the ADSC, the use of a standardized spinal cord contusion model gains scientific strength by controlling several aspects involving the lesion and subsequent evaluations of the possible effects of the administered SC [16]. This paper aims to evaluate the effect of administering human ADSC in the cauda equina of rats submitted to standardized experimental SCI, through histological study of the spinal cord, quantifying cellular changes through immunohistochemistry and histomorphometric analysis; further studying the cellular migration using an in vivo tracking methodology and cell survival rate after transplantation using immunofluorescence.

## 2. Materials and Methods

### 2.1. Study Design

The present study is longitudinal, prospective, randomized, and controlled. It was approved by the Institutional Ethics Committee on the Use of Animals (No. 769) and the Ethics Committee in Research with Humans (CAAE 12723713.3.0000.0020—Universidade Católica do Paraná, Curitiba, Brazil). Three biological samples of adipose tissue donors from bariatric surgery were used. Forty-five Wistar rats, *Rattus norvegicus*, young adults with an average of 20 weeks of age, weighing between 300 and 350 g, female, were used. During the study period, all animals received water and food ad libitum and were kept under light control (light-dark cycles of 12 h) and at room temperature (25 ± 1 °C, continuous temperature monitoring). Spinal cord injury was induced in all animals (n = 45). They were divided into experimental groups (EG, n = 20) and control groups (CG, n = 20). The experimental group was divided according to the number of infusions performed: one infusion after SCI induction (EG1, n = 10) or two infusions, the second infusion seven days after the injury (EG2, n = 10). Control groups received culture medium infusion (CG1, n = 10 and CG2, n = 10). The animals were followed up for 40 days after SCI and euthanized for histological analysis (Figure 1).

### 2.2. Adipose-Derived Stem Cells Isolation and Expansion

Human adipose tissue, obtained through by bariatric surgery, was collected in flasks containing a 2% glucose solution. This tissue was isolated by using enzymatic digestion. Approximately 100 mL of adipose tissue was washed with phosphate buffer saline (PBS) (Gibco™, New York, NY, USA). The tissue was macerated, and enzymatic digestion with collagenase type I (1 mg/mL) (Gibco™, New York, NY, USA) was performed for 30 min at 37 °C, under constant shaking, followed by filtration through 100-μm mesh filter (BD FALCON, BD Biosciences Discovery Labware, Bedford, MA, USA) and centrifugation at 800× *g* for 10 min. Contaminating erythrocytes were removed by erythrocyte lysis buffer. The cells were washed with phosphate buffered saline (PBS) (Gibco™, New York, NY, USA), resuspended in Dulbecco’s modified Eagle’s medium and F12 (DMEM-F12) (Gibco™, New York, NY, USA), supplemented with 10% fetal bovine serum (FBS), penicillin (100 U/mL), and streptomycin (100 µg/mL) (Gibco™, New York, NY, USA), and plated in 75 cm^2^ flasks (TPP, Trasadingen, Switzerland), at a density of 1 × 10^5^ cells/cm^2^. The first change of medium was performed after 24 h, to remove non-adherent cells, and the other changes were performed twice a week. After 80% confluence, cell dissociation was performed with trypsin/EDTA (Gibco™, New York, NY, USA). The culture medium was discarded, and the cells were washed twice with PBS. Then, trypsin was added to the cells and incubated at 37 °C for 4 min. Next, the trypsin was inactivated with 1% FBS and IMDM (Iscove’s Modified Dulbecco’s Medium—Gibco™, New York, NY, USA). The cell suspension was centrifuged, the supernatant discarded, and the cells were counted and plated again (first pass—P1). All experiments were performed between the third (P3) and the fifth passage (P5).

### 2.3. Adipose-Derived Stem Cells Characterization

The characterization of ADSC was carried out by evaluating cell surface markers by immunophenotyping and by cell differentiation potential. The immunophenotypic analysis followed the criteria defined by the International Society for Cellular Therapy (ISCT) [17]. Approximately 1 × 10^6^ cells were placed in each tube, the cells were washed with PBS (Gibco™, New York, NY, USA) and centrifuged for 5 min at 400× *g*. After centrifugation, the following fluorochrome-conjugated monoclonal antibodies were added: CD105, CD90, CD73, CD29, CD44, CD14, CD34, CD45, CD31, and CD19 (BD Pharmingen, San Diego, CA, USA). Next, the cells were incubated with the antibodies for 30 min. They were washed with PBS and resuspended in PBS containing 1% paraformaldehyde (Sigma-Aldrich, Saint Louis, MO, USA). Twenty thousand events were collected from each sample, acquired on a BD FACS Calibur flow cytometer (BD Biosciences), and analyzed using FlowJo software version 10.1 (Tree Star, Ashland, OR, USA).

Cell differentiation potential was assessed using commercial media by inducing ADSC differentiation into adipocytes, osteoblasts, and chondrocytes (Lonza, MD, USA). For adipogenic and osteogenic differentiation, 16,000 ADSC were plated in duplicate on glass coverslips (GlassTécnica, Curitiba, Brazil) in 24-well plates (TPP, Trasadingen, Switzerland). The cells were incubated in an incubator (Thermo Fisher Scientific, Carlsbad, CA, USA) at 37 °C, with 5% CO_2_ until reaching 80% confluence. Hence, 300 µL of differentiation medium (Lonza, Walkersville, MD, USA) was added and the medium was changed three times a week for 21 days. Control cells were cultured for the same period, in a culture medium without differentiation induction factors. For adipogenic differentiation, cells were fixed with Bouin (Biotec, Pinhais, Basil) for 10 min, washed twice with 70% ethanol (Merck, Damstadt, Germany) and with MiliQ water. The cells were stained with a 0.5% Oil Red O solution (Sigma-Aldrich) for 1 h to visualize the presence of lipid vacuoles. Hematoxylin-Eosin (HE) (Biotec) was used for nuclear staining. For osteogenic differentiation, cells were fixed with 1% paraformaldehyde (Sigma-Aldrich) for 20 min and washed with MiliQ ultrapure water. Cells were stained with Alizarin Red S (Fluka Chemie, Steinheim, Germany) for 5 min to assess the presence of calcium crystals.

For chondrogenic differentiation, micromass culture was performed. Briefly, 1 × 10^6^ cells were transferred to a conical tube which was centrifuged for 10 min at 400× *g* to form a pellet. After centrifugation, 1 mL of differentiation medium was added, and medium exchange was performed three times a week for 21 days. Next, control cells were cultured without differentiation induction factors, for the same period. After 21 days, the cell aggregate was fixed with 10% paraformaldehyde (Sigma-Aldrich) for 1 h, dehydrated with serial dilutions of ethanol (Merck) and placed in a paraffin block. Sections of 4 µm thickness were made and stained with toluidine blue (Sigma-Aldrich). The presence of proteoglycans in the extracellular matrix was evaluated using an optical microscope (Eclipse E200, Nikon, Tokyo, Japan) [18].

### 2.4. Adipose-Derived Stem Cells Transduction

The cells were transduced with a luciferase gene for the in vivo monitoring of ADSC by bioluminescence. Cell line Human embryonic kidney 293 cells (HEK 293) was transfected with vectors pMD2.G, pCMV_dr8.91, and pMSCV_Luc2_T2A using Lipofectamine 2000 (Invitrogen, Waltham, MA, USA) and maintained for three days in culture. The supernatant containing the viral particles was collected, filtered with a 0.22-µm filter, and ultracentrifuged at 28,000 rpm for 1 h and 30 min. Cells were resuspended in PBS/BSA 1% (Sigma-Aldrich) and stored at −80 °C. ADSC were transduced with the supernatant containing the viral particles and 10 µg/µL of Hexadimethrine Bromide (Polybrene, Sigma Chemical, St. Louis, MO, USA). The medium used for cell transduction was changed every 24 h for three days. After this period, puromycin (Sigma Chemical, St. Louis, MO, USA) was added for cell selection at a final concentration of 10 mM. The IVIS Lumina II (Caliper Life Sciences, Hopkinton, MA, USA) imaging system was used to analyze the bioluminescence emitted by cells expressing luciferase.

### 2.5. Surgical Procedure

For the induction of standardized experimental SCI, the animals were anesthetized using ketamine (Agener União, São Paulo, Brazil) and 2% xylazine (Agener União, São Paulo, Brazil) intraperitoneally, at a dose of 90 mg/kg and 10 mg/kg weight, respectively. The procedure was started after tests to verify the deep anesthetic plane, absence of corneal reflex, and non-reactivity to caudal compression. Cephalothin 25 mg/kg (Virbac, São Paulo, Brazil) was administered intraperitoneally as preoperative prophylaxis. The animals were positioned in ventral decubitus, on the “Impactor” device (Rutgers University Impactor^®^, Rutgers, NJ, USA), according to a previously validated method of complete SCI through contusion [19]. Trichotomy was performed, and by palpation of the lower ribs, access to the skin and deep dissection at the level of T10 were accomplished, followed by removal of the posterior vertebral elements (laminectomy), exposing the spinal cord. Under controlled and previously validated adjustments of height and speed, a rod weighing 5 g was dropped on the exposed level, generating a complete lesion proven by the specific software (MASCIS^®^, Rutgers, NJ, USA) applied to the system for validation of the lesion. Animals that did not have complete SCI were excluded from the study. After SCI induction, hemostasis was performed, and the surgical wound was washed with 0.9% saline solution, and then the skin was sutured with mono nylon 3.0 thread (Shalom, Goiania, Brazil) (Figure 2). All animals remained under anesthesia for approximately two hours in a chamber with a controlled temperature of 30 °C and placed in individual cages with easily accessible food and water. The surgical wound was inspected daily, and the bladder was emptied using the Credé method every 12 h in all animals throughout the postoperative period [20].

### 2.6. Cell Transplantation

Under anesthesia, a new surgical incision was made, approximately 1.5 cm distal to the first one, identifying the lumbar levels and the dural sac. Under direct visualization, using a Hamilton syringe (Hamilton^®^), a solution of 100 µL of DMEM-F12 medium containing 1 × 10^6^ ADSC was infused inside the dural sac of the animals in groups EG1 and EG2. The same procedure was performed with the animals in CG1 and CG2, with 100 µL of culture medium. The incision was sutured with 3.0 mononylon (Bioline, Anápolis, Goias, Brazil). After seven days of injury and first cell infusion, EG2 and CG2 underwent a new infusion of ADSC or culture medium, respectively.

### 2.7. In Vivo Cell Tracking

The transduced ADSC were infused into three animals from each group. The animals received inhalational anesthesia with isoflurane (Rhodia, Paulínia, Brazil), an intraperitoneal injection of 150 mg/kg of D-luciferin (Perkin Elmer, Waltham, MA, USA) was administered to the animals 5 min before image acquisition. The animals were positioned in the camera in dorsal decubitus and a capture field of 12.7 × 12.7 cm was used (imaging system IVIS LUMINA II, Hopkinton, MA, USA). A series of images were acquired until the maximum peak of the bioluminescence signal was obtained. Images were analyzed using the Living Image software (version 2.50, Xenogen, Caliper Life Sciences, Hopkinton, MA, USA). The signal intensity of the infused ADSC was evaluated 48 h and seven days after the procedure.

### 2.8. Euthanasia and Tissue Collection

Forty days after SCI induction, the animals were euthanized through high-dose anesthesia, intraperitoneally, with ketamine (240 mg/kg weight) (Agener União, São Paulo, Brazil) and xylazine (30 mg/kg weight) (Agener União), associated with halothane (Tanohalo, Cristália, Brazil) gas in an anesthetic chamber (4–5%) to reduce animal suffering, following euthanasia protocols for experimental studies. First, a thoracotomy was performed to allow intracardiac perfusion with 0.9% saline (JP, Ribeirão Preto, Brazil) until complete exsanguination. Then, through the same route, reaching the right atrium, a 4% paraformaldehyde solution was injected (approximate total volume of 300 mL/animal) (Neon, Suzano, Brazil), allowing the immediate fixation of the nervous tissue and providing ideal conditions for preparation and anatomopathological manipulation.

### 2.9. Anatomopathological Study—Histology and Immunohistochemistry

After euthanasia, the spinal cord was collected and transferred to a 10% formalin buffered solution. After fixation, histological sections were made, with a thickness of 5 µm from the central area of the lesion. Evident morphological changes were verified macroscopically, evidencing opaque and darker coloration, as well as local thinning, in correspondence with the experimentally induced SCI level (Figure 3).

For histological analysis, samples were stained with hematoxylin-eosin (HE), luxol fast blue (Sigma-Aldrich) and cresyl violet (Sigma-Aldrich). From the analysis of the slides stained with luxol fast blue, a software, Image ProPlus (Media Cybernetics, Silver Spring, MD, USA), based on the color morphometry method, estimated the amount of myelin present in the tissue. Using cresyl violet staining, two independent examiners assessed medullary cellularity, focusing on neuronal presence, by dividing the histological field into equal quadrants (anterior-posterior, right-left, and combinations), using an optical microscope at 200× magnification (Olympus, Tokyo, Japan). Eventual neurons present in medullary sections were counted separately. The software calculated the marked area in units of square micrometers (µm^2^). Some values were expressed in the total value of points identified by the program, to better demonstrate the differences between groups [21]. Glial fibrillary acidic protein (GFAP) was evaluated by immunohistochemistry technique to identify activated astrocytes.

### 2.10. Statistical Analysis

Data analysis was performed using the SPSS Statistics software (v. 20.0). In the cases of histological and immunohistochemical evaluations, the means were compared, and the non-parametric Mann–Whitney test was applied with significance level corrected by Bonferroni, in addition to the one-way analysis of variance (ANOVA, *p* = 0.05). Values *p* ≤ 0.012 were considered significant. Student’s *t* test was used for comparison within groups (*p* < 0.05). The data obtained from the in vivo tracking of the cells were not submitted to statistical analysis, being presented in a descriptively way.

## 3. Results

### 3.1. Animal Care and Surgical Procedures

The animal model of SCI induction was adequate, with observations as follows. Five animals were excluded from the study: four died, three during the anesthetic period of induction and prolonged maintenance to perform the surgery and evaluate cell tracking in vivo, probably due to idiosyncratic reactions; and one during the late maintenance period, probably related to complications arising from and inherent to the loss of bladder control and neurological function. One animal did not present complete spinal cord injury and was excluded.

### 3.2. ADSC Characterization

The ADSC surface antigen profile was evaluated, and they were accordingly the criteria defined by ISCT [19]. The cells were positive for CD105, CD90, CD73, CD29, and CD166 markers and negative for CD34, CD45, CD19, CD14, and HLA-DR. Cellular viability before injection into the spinal canal was 98.2% (Figure 4).

After induction of cell differentiation with specific media for each lineage, ADSC showed differentiative potential into adipocytes, osteoblasts, and chondrocytes (Figure 4).

### 3.3. Histomophometric Analysis of Myelin and Neurons

Adipose-derived stem cell infusion did not change the myelin area significantly in comparison with the control group. In three samples, tissue destruction was extremely intense such that histological analysis was impossible. There was a large variation in the amount of myelin measured within the groups (Figure 5).

There was a significant difference regarding number of neurons in the injury area when comparing the experimental and control groups. Groups that received ADSC infusion showed significantly more neurons. There was no difference when comparing one and two ADSC infusions (Figure 6).

### 3.4. Immunohistochemical Astrocyte Evaluation

The spinal cord was divided in four sections (anterior right and left, posterior right and left). There was no significant difference when comparing experimental and control groups (Table 1 and Table 2, Figure 7).

### 3.5. In Vivo Cell Tracking with Bioluminescence Imaging

Bioluminescence evaluation was performed in three animals of each experimental group, at 48 h and seven days after first ADSC infusion (EG1 and EG2). In EG2, the evaluation was performed 48 h and seven days after second cell infusion. At 48 h of evaluation, cells were observed at the injury site and in some animals on the lungs. It was noticed that signal intensity was reduced on the second evaluation for EG1 (day 7) and EG2 group (days 7 and 9) after infusion. In the animals of the groups EG1 and EG2, no cells were observed after seven days of ADSC infusion (Figure 8).

## 4. Discussion

The methodology used in this study, standardized spinal cord contusion, aims to allow for better comparison within our own institution and with other research centers globally. In comparison with clip or balloon compression, the impactor device allows for better similarity in histology and motor recovery and leads to fewer deaths during follow-up [22,23].

The choice of ADSC followed not only our institution’s protocol but is also justified by benefits found in other similar studies: good resistance to hypoxic conditions, secretion of neurotrophic factors, and low immunogenicity. It is possible that the combination of hypoxic conditioning and the addition of different cell types will significantly improve the regenerative potential of ADSC [24]. Furthermore, these cells resist hypoxic conditions, secrete neurotrophic factors, and show low immunogenicity [25]. The injection pathway was the main modification in our research strategy, due to unsatisfactory results achieved with direct injection into the injury site, especially in the acute setting. Our previous reports show decreased motricity after direct stem cell injection into the injury site [10]. Although there are reports of cellular migration towards the injury with intravenous or arterial catheterization pathways, cellular concentration is higher when the cells are delivered directly into the nervous system [26]. In the acute setting, the inflammatory process makes it challenging to identify the presence of stem cells, and in the chronic phase, fibrosis limits tissue expansion, limiting the injected volume and presenting evident extravasation from the injury site [27]. As proposed by our protocol, lumbar punction is also accessible for human trials, and understanding the behavior of stem cells injected into this pathway is crucial for developing future human therapeutic strategies.

In vivo, cellular tracking with bioluminescence imaging is a sensitive technique for evaluating administered cell distribution, localization, and migration [28]. Although ADSC have a short life span after infusion [29], this research identified cells at the injury site and on the lungs right after each infusion. Cell tracking with bioluminescence allowed us to observe the migratory capacity of ADSC towards the lesion site; however, the animals that received repeated sedation during the injury recovery period showed a significant mortality rate. Repeated anesthetic procedures can also be a negative factor in evaluating a modulation and cell survival. However, in this study, no significant histological differences were noticed between animals that underwent cell tracking and those that did not. Callera et al. [30] were able to track marked cells using magnetic resonance imaging, confirming migration from the infusion site in the lumbar spine towards the injured area in the thoracic spinal cord. Other methods to track cell migration include magnetic resonance imaging, in vivo microscopy, and micro-computed tomography. However, these methods require expensive equipment, making results reproducibility difficult, especially in low-income countries [31]. The myelin sheath, produced by oligodendrocytes in the central nervous system, renders the structure to the spinal cord and regulates neuronal growth through neurite outgrowth inhibitor (NOGO) molecules secreted by injured or dysfunctional oligodendrocytes. Controlling myelin production and NOGO secretion is critical to spinal cord regeneration research [32,33]. In the present study, ADSC could not effectively modify the structural damage secondary to myelin loss nor activate astrocytes. On the contrary, some studies showed that mesenchymal stromal cell promotes axonal regeneration and myelination [34]. Chen et al. reported that the infusion of ADSC directly to the injury site right after the trauma promoted astrocyte activation and reduced glial scar [25].

In this research, neuronal loss was reduced, which suggests that immunomodulatory factors secreted by the ADSC reduced inflammation, inhibited apoptosis, and protected neurons. These results are in line with other studies showing that the intravenous injection of ADSC had a similar effect on the number of neurons, even with a small concentration of cells migrating towards the injury site on subacute phases of SCI, which suggests a regenerative effect secondary to ADSC administration [35]. Menezes et al. identified laminin deposition on the injury site, indicating neuronal regeneration after cell transplantation [36].

Another key issue in spinal cord regeneration is controlling scar formation. By not controlling or reducing astrocyte migration to the injury area, ADSC could not block the intrinsic process that limits axonal regrowth and neuronal reconnection [37].

The infusion of cells into the dural sac, distal to the spinal cord, allows for a larger injected volume, with low risk of neurologic deterioration, and in vivo cell tracking confirmed cell migration towards the injury site. The downside of repeated anesthetic procedures in the experimental models, would not be a problem on a human trial, since lumbar puncture can be performed on an ambulatory fashion, with local anesthesia.

## 5. Conclusions

ADSC infusion distal to the injury site was able to reduce neuronal loss significantly, although it did not prevent the myelin loss or enhance the area occupied by astrocytes compared to the control group. The results are similar when comparing one or two cell infusions.

## Figures and Tables

**Figure 1 biomedicines-11-01394-f001:**
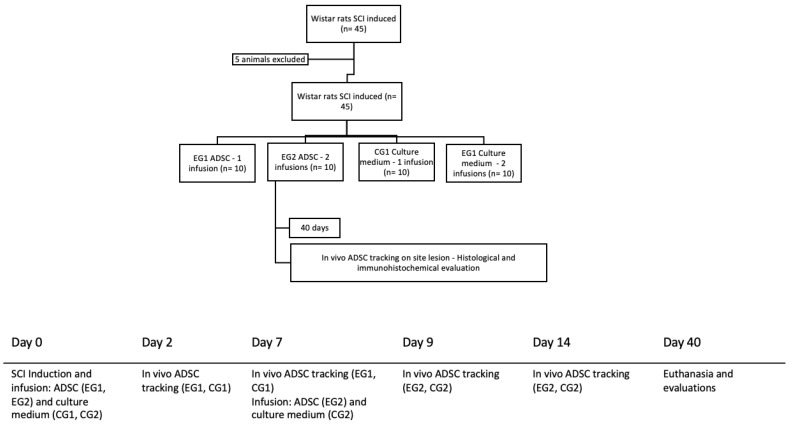
Flow diagram of experimental study. SCI (spinal cord injury); EG (experimental group); ADSC (adipose derived stem cells); CG (control group).

**Figure 2 biomedicines-11-01394-f002:**
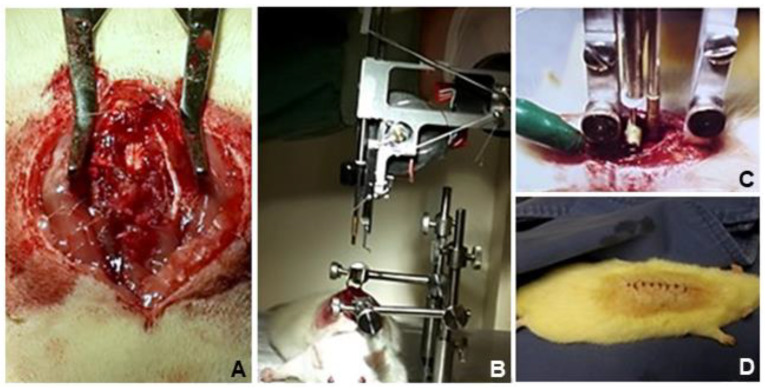
Spinal cord injury procedure. (**A**). Dissection of T9-T10 level followed by laminectomy, exposing the spinal cord. (**B**). Animal positioned on the Impactor device. (**C**). A rod was dropped on the spinal cord, generating a lesion. (**D**). After the procedure, the skin was sutured.

**Figure 3 biomedicines-11-01394-f003:**
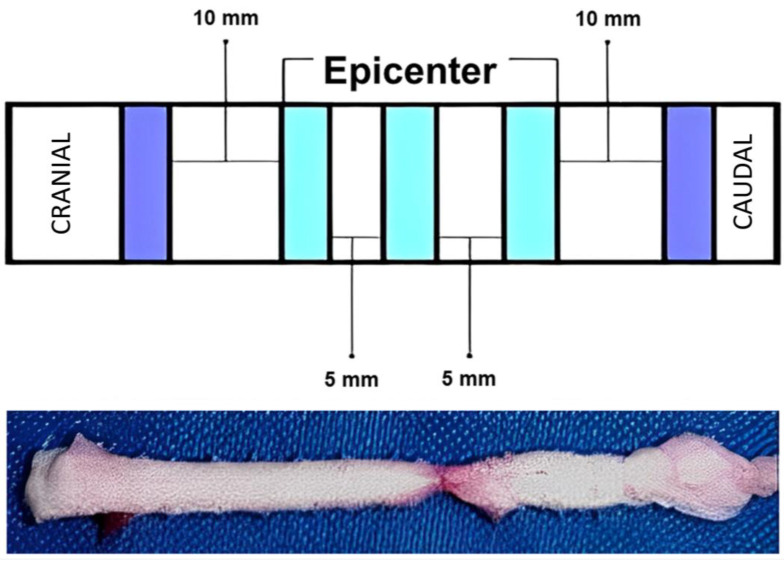
Anatomopathological image of the spinal cord. After removal of the surrounding bone, the tissue was fixed and prepared for histological sections. The central cuts were 5 µm thick, with 5 mm intervals for the central area and 10 mm for the distal cuts. The blue areas had 6 cuts and the violet ones had three cuts.

**Figure 4 biomedicines-11-01394-f004:**
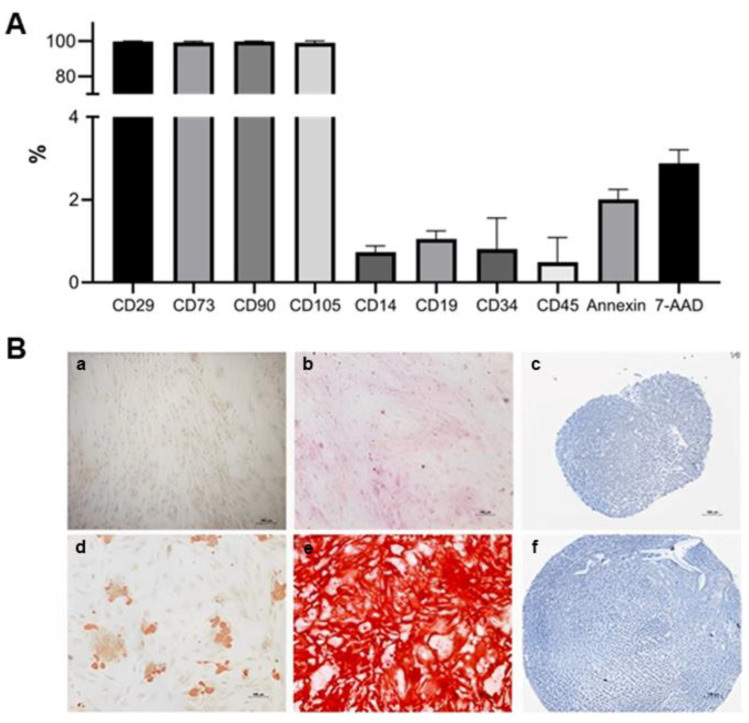
Adipose-derived stem cell characterization. (**A**). Immunophenotypical profile, cell viability, and apoptosis/necrosis. The bars represent the average expression for each marker. (**B**). Adipose-derived stem cell differentiation. Cells were maintained with medium to induce differentiation for 21 days. (**d**). Adipose differentiation was demonstrated with the presence of lipid vacuoles. (**e**). Extracellular matrix mineralization was observed at osteogenic differentiation. (**f**). Chondrocytic differentiation was observed by proteoglycan deposition in the matrix and the presence of lacunae around young chondrocytes. (**a**–**c**). Control cells were cultured without an induction medium.

**Figure 5 biomedicines-11-01394-f005:**
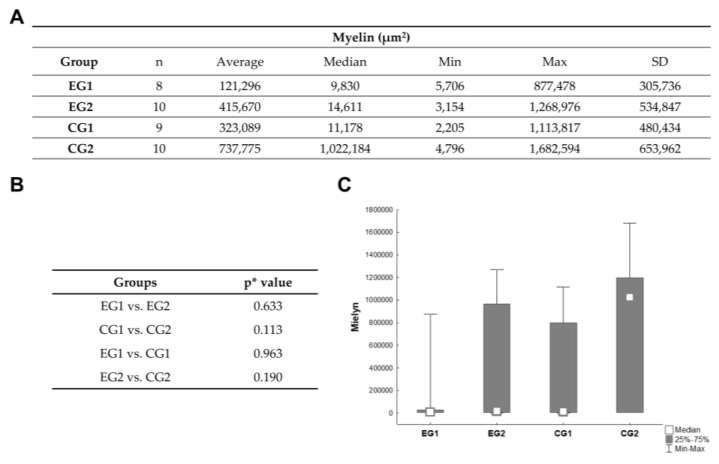
Histomorphometric analysis of myelin area. (**A**,**B**). Comparison of myelin area showed no significant difference between groups. (**C**). Myelin was measured in units of µm^2^ and represented by the median values. * Mann-Whitney non-parametric test, *p* < 0.012 (Bonferroni’s correction). EG: experimental group; CG: control group; SD: standard deviation.

**Figure 6 biomedicines-11-01394-f006:**
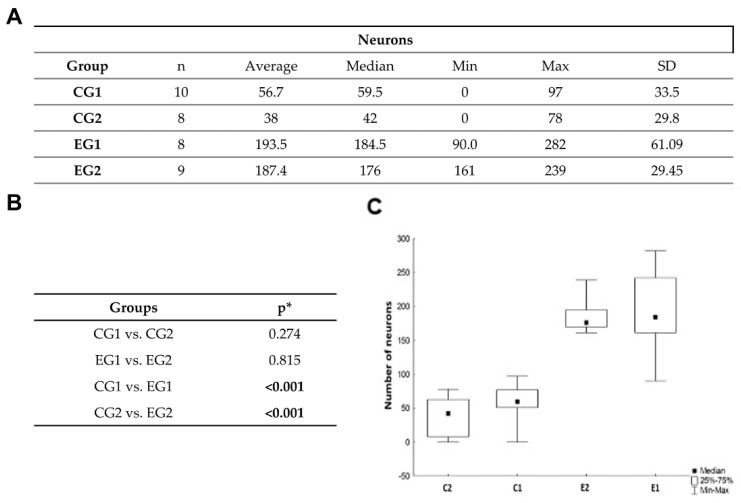
(**A**) Histomorphometric analysis of neurons. (**B**,**C**) Comparison of neuron numbers between experimental and control groups. Groups that received ADSC infusion, showed a greater number of neurons. * Mann-Whitney’s non-parametric test, *p* < 0.012 (Bonferroni’s correction). EG: experimental group; CG: control group; SD: standard deviation. Numbers in bold represent statistically significant difference.

**Figure 7 biomedicines-11-01394-f007:**
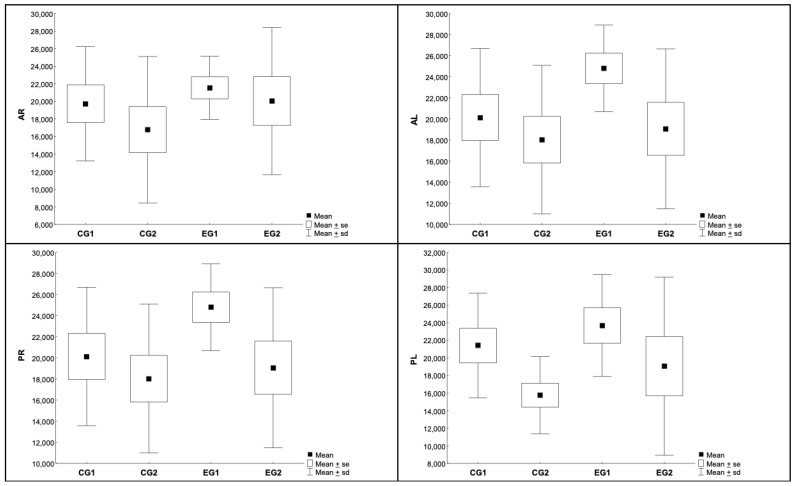
Immunohistochemical evaluation of astrocytic area, divided in four quadrants—EG: experimental group; CG: control group; AR: anterior-right; AL: anterior-left; PR: posterior-right; PL: posterior-left.

**Figure 8 biomedicines-11-01394-f008:**
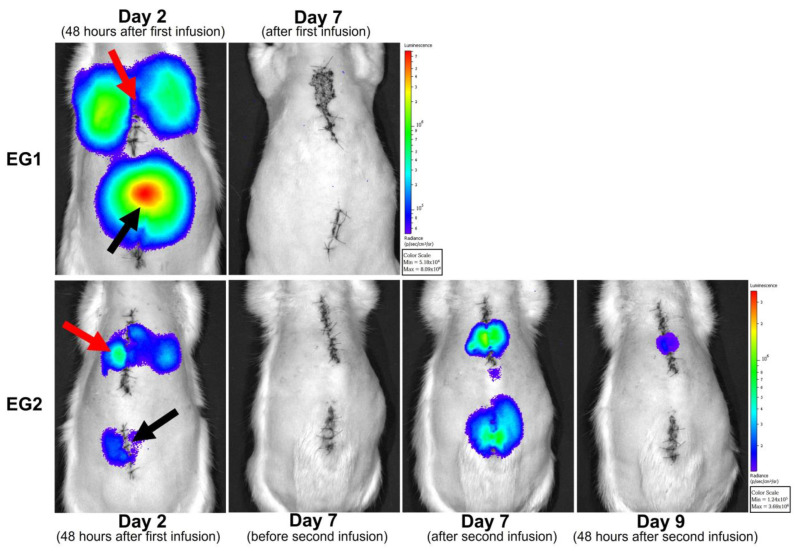
In vivo tracking of ADSC. Representative photos of experimental groups (EG1 and EG2). Cells were observed at the infusion site and in the lungs. The red arrow indicates the injury site, and the black arrow indicates the ADSC infusion site. The images are pseudo-colored and represent the bioluminescent signal emitted during degradation of D-luciferin by the enzyme luciferase present in cells. The color scale represents the intensity of the bioluminescence signal emitted by the cells in photons/second/cm^2^/steradian (p/s/cm^2^/sr).

**Table 1 biomedicines-11-01394-t001:** Description of astrocytic area in four quadrants. Astrocytes were measured in units of µm^2^. Analysis of variance with one factor (ANOVA), *p* < 0.05—EG: experimental group; CG: control group; AR: anterior-right; AL: anterior-left; PR: posterior-right; PL: posterior-left.

Area	Group	N	Average	Median	Min	Max	SD
AR	CG1	9	19,738	18,760	8690	28,785	6500
	CG2	10	16,778	15,287	7870	36,177	8326
	EG1	8	21,547	21,702	15,805	26,511	3597
	EG2	9	20,042	17,650	10,312	34,553	8369
AL	CG1	9	19,882	21,492	7922	27,080	6766
	CG2	10	13,586	12,688	5525	25,587	6325
	EG1	8	21,256	20,114	15,863	29,074	4339
	EG2	9	18,629	18,453	9779	37,710	9531
PR	CG1	9	20,139	20,121	8302	30,488	6559
	CG2	10	18,034	16,628	8737	32,206	7045
	EG1	8	24,795	24,927	15,770	29,270	4107
	EG2	9	19,064	21,752	10,207	31,100	7589
PL	CG1	9	21,411	21,664	10,541	29,412	5929
	CG2	10	15,765	16,124	8161	22,956	4404
	EG1	8	23,672	23,148	16,040	31,780	5776
	EG2	9	19,058	13,830	11,131	38,737	10,120

**Table 2 biomedicines-11-01394-t002:** Comparison of astrocytic area between groups. Student’s *t*-test for independent samples, * *p* < 0.012 (Bonferroni’s correction); ANOVA, *p* < 0.05—EG: experimental group; CG: control group; AR: anterior-right; AL: anterior-left; PR: posterior-right; PL: posterior-left.

*p* * Value				
Groups	AR	AL	PR	PL
EG1 × EG2	0.633	0.486	0.077	0.275
CG1 × CG2	0.404	0.051	0.511	0.033
EG1 × CG1	0.497	0.631	0.105	0.439
EG2 × CG2	0.407	0.188	0.763	0.387

## Data Availability

Data is available upon request to the main author, according to Institutional policy.

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
