# Peer review of "Human Adipose-Derived Stem Cells Reduce Cellular Damage after Experimental Spinal Cord Injury in Rats"

_biomedicines, 2023, doi:10.3390/biomedicines11051394_

Round 1
Reviewer 1 Report
This article showed that human adipose-derived stem cells reduce cellular damage after experimental spinal cord injury in rats. There are some concerns in this manuscript that should be addressed as follows:
1. Abstract Line 18: The meaning of the abbreviation "GFP" should be mentioned.
2. A conclusive statement should be added to the abstract.
3. The novel points in this study should be clarified because there are previous studies that discussed a similar topic.
4. The nation and the institution to which the Ethics Committee that approved this study belongs should be mentioned.
5. The source of animals used in this study should be mentioned.
6. How did the authors know that the animals were acclimatized?
7. Why did the authors selected values p≤0.012 to be considered significant, although most literature considered values less than 0.05 to be statistically significant.
8. A collective diagram that summarizes the main findings of the present study should be added.
9. The discussion should be rewritten to focus on analysis of the results of the present study.
10. The limitations of the present study should be mentioned.
11. The conclusion should be rewritten to be in the form of a paragraph that summarizes the main findings of the present study and describes the clinical implications of the findings of the present study.
12. The number of references is relatively small for a research article. Also, more recent references should be added.
13. The manuscript should be checked regarding the grammatical and typing errors.
Author Response
Thank you for your time and effort to help improving our manuscript. We have adapted the text as requested. All changes are highlighted on the reviewed manuscript and listed bellow
- Abstract Line 18: The meaning of the abbreviation "GFP" should be mentioned. Done
- A conclusive statement should be added to the abstract. Done
- The novel points in this study should be clarified because there are previous studies that discussed a similar topic. A paragraph was added to the discussion regarding new points
- The nation and the institution to which the Ethics Committee that approved this study belongs should be mentioned. Done
- The source of animals used in this study should be mentioned. Done
- How did the authors know that the animals were acclimatized? this was added to the methodology as requested
- Why did the authors selected values p≤0.012 to be considered significant, although most literature considered values less than 0.05 to be statistically significant. We use Bonferroni's test for small samples, which is more precise than Student's test, and uses p<0,012. When Student's test was used , the value was 0.05
- A collective diagram that summarizes the main findings of the present study should be added. A diagram was added to the results
- The discussion should be rewritten to focus on analysis of the results of the present study. We have rearranged the discussion according to your request
- The limitations of the present study should be mentioned. the final paragraph now focus on the limitations.
- The conclusion should be rewritten to be in the form of a paragraph that summarizes the main findings of the present study and describes the clinical implications of the findings of the present study. conclusion was formatted according to your request
- The number of references is relatively small for a research article. Also, more recent references should be added. References were updated and revised as requested.
- The manuscript should be checked regarding the grammatical and typing errors. Done

Reviewer 2 Report
In the manuscript entitled “Human Adipose-Derived Stem Cells Reduce Cellular Damage After Experimental Spinal Cord Injury in Rats” the authors show that ADSCs infused inside the dural sac migrated towards a spinal cord injury site. There, the number of neurons was significantly increased in comparison with a control group 40 days after the injury. However, myelin area and astrocyte number were not increased.
ADSCs will become increasingly important in regenerative medicine and therefore the topic is of great interest and importance. The study is well conducted. However, some concerns have to be addressed. The language needs careful revision. Most of the figures are of poor quality and need a higher resolution. Also, the authors should revise their graphics, it seems that the values do not the match the values from the corresponding tables (unless there is a misunderstanding from my side).
Minor concerns
· lines 41/42: “are presented so far without as an ineffective option in the treatment of SCI”
Please rephrase, I am not sure what You are trying to say here (“without as an ineffective option”?)
· lines 45/:46 “tissue regeneration. and functional, with possible positive clinical correlation”
Please remove the period after “regeneration”. Is there a word missing after “functional”? Please rephrase the sentence.
· lines 63/64: “After approval by the Institutional Ethics Committee on the Use of Animals (nº 769) of the as well as by the Ethics Committee”
There seems to be a word missing (“of the as well as”?)
· Actually, the whole manuscript needs a careful revision on the language. I will stop here to point out the individual language mistakes. Please revise the language of the manuscript.
· lines 77/78: “The animals were followed up for 40 days after the last procedure they underwent, being euthanized for sequent analysis”
This implies, that the euthanasia of groups EG2 and CG2 would habe been one week later than the euthanasia of groups EG1 and CG1. The organogram in Figure 1 implies that the euthanasia of all groups was at the same time. Please clarify.
· line 124: please explain the abbreviation “IMDM”
· line 292: is “ADST” supposed to be “ADSC”?
· I don’t really understand the labeling of the pictures in “figures” and “graphics”. I see that the figures 2, 3, 5, 6, 7 are photographs, and graphics 1 to 3 are diagrams. But why is figure 4 a figure and not a graphic?
· Graphic 1: the name of the y-axis is distorted and covers the axis labeling. Are the values for graphic 1 taken from table 1? If so, the medians for EG1, CG1 and EG2 seem wrong. Also, the black bars are supposed to show the 25%-75% but all start at zero? Is this correct? I am not sure what I am looking at, here.
· I think there is a mistake in table 4. In the third row, it should read “EG1 x CG1”, not “CG1 x CG1”?
· Is the figure legend of graphic 2 in portuguese (“Mediana”)?
· Are the values for graphic 2 taken from table 3? If so, the medians seem wrong. In table 3, the median of CG2 is lower than the median for CG1. However, in graphic 2 it’s the other way.
· Graphic 3 has a too low resolution. Actually, all graphics are of inferior quality, but the y-axis labeling of graphic 3 is unreadable. Also, the legend in figure 7 is unreadable.
· lines 348/349: “Signal intensity was reduced on the second evaluation (day 7)”. Well, it seems for EG1 there is no signal at all on the second evaluation. Also: was the picture for EG2 taken before the second cell infusion? Please clarify!
· lines 370-372: “The injection pathway was the main modification in our research strategy, due to the unsatisfactory results achieved with direct injection into the injury site, especially in the acute setting.” Are these unpublished results? Or are You referencing something here? Please clarify!
· lines 419/420: “There was no difference between one and two ADSC injections on all histological analysis.” The plural of “analysis” is “analyses”. However: is this statement correct? In graphic 3, it seems that the difference between one and two treatments (regardless of experimental or control group) is much bigger than the difference between experimental and control group?
Author Response
Minor concerns
- lines 41/42: “are presented so far without as an ineffective option in the treatment of SCI”
Please rephrase, I am not sure what You are trying to say here (“without as an ineffective option”?)
Authors agree and say sorry because of the misunderstood; text corrected: “other options involving active molecules (corticoids, riluzole, hormones, calcium channel blockers, and specific proteins) are presented so far as ineffective in the treatment of SCI [8,9].”
- lines 45/:46 “tissue regeneration. and functional, with possible positive clinical correlation”
Please remove the period after “regeneration”. – authors agree and inform that text was corrected.
3.Is there a word missing after “functional”? Please rephrase the sentence. – authors agree and inform the corrected sentence as follow “The application of stem cells (SC) presents itself as a promising option in the face of this reality, since it is potentially capable of both interrupting or minimizing the cascade of pathophysiological events and providing a favorable environment for specialized tissue regeneration and local functionalization, with possible positive clinical correlation [10-12].”
- 4. lines 63/64: “After approval by the Institutional Ethics Committee on the Use of Animals (nº 769) of the as well as by the Ethics Committee”
There seems to be a word missing (“of the as well as”?) - authors agree and inform the corrected sentence as follow “After approval by the Institutional Ethics Committee on the Use of Animals (nº 769) as well as by the Ethics Committee in Research with Humans (CAAE 12723713.3.0000.0020), the experimental study was started.”
The whole manuscript needs a careful revision on the language. I will stop here to point out the individual language mistakes. Please revise the language of the manuscript – OK, authors thank for the observation and inform revision done.
- 5. lines 77/78: “The animals were followed up for 40 days after the last procedure they underwent, being euthanized for sequent analysis”
This implies, that the euthanasia of groups EG2 and CG2 would have been one week later than the euthanasia of groups EG1 and CG1. The organogram in Figure 1 implies that the euthanasia of all groups was at the same time. Please clarify.
Authors corrected the text, hoping to clarify better the idea, as follows “The animals were followed up for next 40 days since the SCI induction procedure they have been undergone, being euthanized for sequence analysis (Figure 1).”
- 6. line 124: please explain the abbreviation “IMDM”
Authors explained, as follow: “IMDM (Iscove's Modified Dulbecco's Medium – GibcoTM, NY USA)”.
- 7. line 292: is “ADST” supposed to be “ADSC”?
Authors feel sorry for the mistake; correction done for “ADSC”.
- 8. I don’t really understand the labeling of the pictures in “figures” and “graphics”. I see that the figures 2, 3, 5, 6, 7 are photographs, and graphics 1 to 3 are diagrams. But why is figure 4 a figure and not a graphic?
Authors would like to inform that this sequence is corrected; Figure 4 was renamed as Graphic 1, changing all the numbers sequence in accordance.
- 9. Graphic 1: the name of the y-axis is distorted and covers the axis labeling. Are the values for graphic 1 taken from table 1? If so, the medians for EG1, CG1 and EG2 seem wrong. Also, the black bars are supposed to show the 25%-75% but all start at zero? Is this correct? I am not sure what I am looking at, here.
The mean values were so divergent that a graphic representation was not possible so we have chosen to present the graphic with the median values. The values were reviewed with the statistician.
- 10. think there is a mistake in table 4. In the third row, it should read “EG1 x CG1”, not “CG1 x CG1”?
Third row corrected for EG1 x CG1.
- 11. Is the figure legend of graphic 2 in portuguese (“Mediana”)?
Text corrected.
- 12. Are the values for graphic 2 taken from table 3? If so, the medians seem wrong. In table 3, the median of CG2 is lower than the median for CG1. However, in graphic 2 it’s the other way.
Data is adjusted. Thank you for pointing that out.
- 13. Graphic 3 has a too low resolution. Actually, all graphics are of inferior quality, but the y-axis labeling of graphic 3 is unreadable. Also, the legend in figure 7 is unreadable.
All graphics were adjusted.
- 14. Lines 348/349: “Signal intensity was reduced on the second evaluation (day 7)”. Well, it seems for EG1 there is no signal at all on the second evaluation. Also: was the picture for EG2 taken before the second cell infusion? Please clarify!
A better description was added to the text.
- 15. Lines 370-372: “The injection pathway was the main modification in our research strategy, due to the unsatisfactory results achieved with direct injection into the injury site, especially in the acute setting.” Are these unpublished results? Or are You referencing something here? Please clarify!
A reference was added from our previous work.
- 16. Lines 419/420: “There was no difference between one and two ADSC injections on all histological analysis.” The plural of “analysis” is “analyses”. However: is this statement correct? In graphic 3, it seems that the difference between one and two treatments (regardless of experimental or control group) is much bigger than the difference between experimental and control group?
All the items in that section were reviewed. The variation on graphic 3 (now graphic 4) was huge, leading to no statistical difference. The histological analysis of astrocytes is challenging due to the necrosis on the central part of the injured spinal cord.

Round 2
Reviewer 1 Report
The authors had appropriately addressed most of my comments
Author Response
thank you, we have further reviewed grammar and typing errors. the quality of the images was adjusted, and reference numbers verified and corrected